# Temporal responses in sensorimotor cortex during hand movements

**Sophia Gimple** [1,2], **Zachary V. Freudenburg**[1], **Nick F. Ramsey**[1,3], **Mariana P. Branco** [1,3]*

**1** Department of Neurology and Neurosurgery, UMC Utrecht Brain Center, Utrecht University, Utrecht, The Netherlands, **2** Department of Neurosurgery, Mental Health and Neuroscience Institute, Faculty Health, Medicine and Life Sciences, Maastricht University, Maastricht, The Netherlands, **3** Donders Institute for Brain, Cognition and Behaviour, Radboud University, Nijmegen, The Netherlands

* m.pedrosobranco@umcutrecht.nl

## Abstract

The role of primary somatosensory cortex (S1) in sensorimotor integration, classically limited to sensory processing, has recently been challenged. In particular, it is unclear whether neural signals in S1 activate prior to or after the primary motor cortex (M1) in the presence and absence of reafferent feedback. Here we aim to gain a better understanding of the temporal dynamics between S1 and M1, and the underlying neural processes involved in movement execution. We compare the onset latency of S1 and M1 signals in eight able-bodied participants implanted with high-density electrocorticography (ECoG) grids over the hand region of the sensorimotor cortex for both low- and high-frequency band (LFB and HFB) signals. Our results show a consistent activation of M1 electrodes before S1 electrodes across able-bodied participants for the HFB but no clear pattern for the LFB. Furthermore, we compared these results with two participants implanted with ECoG, who attempted movement with no or minimal reafference, showing similar effect in the HFB signals. These results call for an updated view of S1 beyond processing of sensory input, as they suggest S1 may also play a role during movement initiation. Better understanding of the integration of M1 and S1 signals will be undoubtedly critical for the understanding of neuromuscular disorders and the development of neurotechnology.

## 1. Introduction

The human hand holds an intricate anatomy that enables the performance of a variety of functions from controlling objects to communication [1–3]. Undoubtedly, the coordination of these functions demands a large and detailed representation of the hand in the brain. The cortical representation of hand movements is centrally located in the primary motor cortex (M1, Brodmann area 4) and the primary somatosensory cortex (S1, Brodmann area 1–3) [4,5]. The exact role of M1 and S1 cortices during movement is not trivial, with early reports from Penfield and Boldrey presenting

**Data availability statement:** The data that support the findings of this study are available on reasonable request from the corresponding author. The data are not publicly available due to privacy restrictions. For data request please conact our data manager Mariska Vansteensel (m.j.vansteensel@umcutrecht.nl).

**Funding:** This research was funded by the Dutch Research Council (NWO): NeuroCIMT project (grant 14906, NFR) and PANDA project (grant 19072, MPB).

**Competing interests:** The authors declare no conflicts of interest.

sensation and movement on both sides of the central sulcus, therefore indicating a complex communication network between these two regions [6].

While both areas are known to be involved in movement initiation, execution and control [7], M1 is typically associated with motor output [8], whereas S1 is mainly associated with processing afferent somatosensory feedback (e.g., [9]). In the last decades, however, a more multifaceted view of S1's role in sensorimotor integration has emerged [10]. Evidence shows that S1 is able to drive movements independently from M1 [11], and is directly linked to motor learning (e.g., [12]) and movement planning [13] through cortical feedforward mechanisms that predict and integrate efference copies of motor commands with incoming sensory signals [14].

To better understand the complex interplay between M1 and S1, several studies aimed to describe the pattern of flow of information between M1 and S1 both in terms of direction and amplitude. In particular, recent studies using electrocorticography (ECoG) in human and non-human primates have attempted to quantify the direction information flow and its relation to movement initiation [15–17], albeit with contradictory results. While Sun and colleagues report that high-frequency band signals in S1 electrodes increased before those in M1 electrodes in humans [16], Witham's and Umeda's studies reported the opposite [15,17], suggesting that S1 received motor-initiated neural signals from M1. Notwithstanding this contradiction, these studies consistently support the idea that S1 activates before movement onset either by receiving or sending information from/to M1 before receiving sensory input. This finding is further supported by multiple studies who demonstrated that hand movements can also be well discriminated from S1 alone in people with upper limb amputation [18–20] or with severe paralysis [21–23], for whom sensory input is absent. Indeed, previous studies have described a large overlap of brain activation patterns in executed and attempted movement indicating a similarity in underlying processes, including activity in sensory regions in amputees [24,25].

To address the inconsistent findings on timing, we analyzed the onset latency of neural activity in M1 and S1 using ECoG, in relation to movement onset. To separate somatosensory feedback from feedforward mechanisms to S1, we included two participants with limited/no ability to move their hand who were implanted with electrodes on the affected hand region. Their movement attempt should activate the M1 but does result in minimal or no feedback input to S1 due to amputation or paralysis. We focus on the high-frequency band power (60–100 Hz) given this is correlated with firing rates in neuronal populations as measured with ECoG [26,27] but include analyses of lower bands (8–30 Hz) due to their role in motor planning [28]. We expect electrodes over M1 to show earlier neural activation than those over S1, with both regions activating before movement onset.

## 2.  Materials and methods

### Participants

Eight able-bodied participants were included in this study (P01-P08; 6 females, age range 19–50, mean age 34.63 ± 11.31; Table 1). Participants were implanted with subdural ECoG grids for the localization of seizure onset in intractable

Table 1. Participant overview. Overview of participants' age (at time of recording), gender, handedness, moved hand (hand), implanted hemisphere (hemisphere), grid manufacturer, inter-electrode distance (IED), exposed electrode diameter (diameter), number of channels in CAR (# ch in CAR), number of included channels over M1 (# M1 ch) and number included of channels over S1 (# S1 ch). Black vertical line separates the participants who executed movement from those who attempted movement.

| Participant ID | 01 | 02 | 03 | 04 | 05 | 06 | 07 | 08 | 09 | 10 |
|---|---|---|---|---|---|---|---|---|---|---|
| Age | 43 | 19 | 45 | 34 | 20 | 36 | 50 | 30 | 63 | 58 |
| Gender | M | F | F | M | F | F | F | F | M | F |
| Handedness | R | R | L | R | R | R | L | R | unknown | R |
| Hand | R | R | L | L | R | R | L | L | R | R |
| Hemisphere | L | L | R | R | L | L | R | R | L | L |
| Grid Manufacturer | AdTech | AdTech | AdTech | PMT | PMT | AdTech | PMT | PMT | AdTech | Medtronic |
| IED (mm) | 3 | 3 | 3 | 4 | 4 | 4 | 3 | 3 | 5 | 10 |
| Diameter (mm) | 1.3 | 1.3 | 1.3 | 1 | 1 | 1.17 | 1 | 1 | 2.3 | 2.3 |
| # ch in CAR | 112/128 | 113/128 | 117/128 | 64/64 | 63/64 | 128/128 | 120/128 | 123/128 | 60/63 | 16/16 |
| # M1 ch | 10 | 13 | 6 | 18 | 16 | 26 | 28 | 36 | 19 | 3 |
| # S1 ch | 5 | 12 | 37 | 10 | 18 | 16 | 25 | 7 | 18 | 4 |

epilepsy at the University Medical Center (UMC) Utrecht, and agreed to the implant of an extra high-density (HD) grid for research purposes. HD grids had an inter-electrode distance of 3 or 4 mm and an exposed diameter of 1 to 1.3 mm (AdTech, Racine, USA; or PMT Corporation, Chanassen, MN, USA) and consisted of 32, 64 or 128 channels located above the hand-knob region of the sensorimotor cortex covering both the M1 and the S1 (Fig 1). In this study, we focused our analysis on the data collected with HD grids. In addition, data from two participants who performed attempted movement due to motor-impairments were also included. P09 was a 63-year-old, male patient with right hand amputation who was implanted with an epidural ECoG grid for evaluation of pain treatment at the University of Tübingen (see [29] for details). The ECoG grid included 64 channels with 2.3 mm exposed diameter and 5 mm inter-electrode distance (AdTech, Racine, USA; or PMT Corporation, Chanassen, MN, USA). Coverage included the hand-knob area of the sensorimotor cortex. P10 was a 53-year-old female with locked-in syndrome due to late-stage Amyotrophic Lateral Sclerosis (ALS), implanted with subdural Medtronic 4-electrode strips over the hand region of the sensorimotor cortex (Resume II from Medtronic with 4 mm exposed diameter and 10 mm inter-electrode distance; see [30] for details). P10 had an ALS functional rating scale score of 2 with no visible reafference, as measured with electromyography on the hand and arm.

Data from participants P01-P08 and P10 were collected under two protocols approved by the Medical Ethical Committee (NedMec) of UMC Utrecht. P01–P08 provided written informed consent for participation in the original study and for future data reuse. P10 gave verbal informed consent for the original study and future data reuse through a tailored procedure described in [30]. P09 provided informed consent for the original study and data publication under approval from the local ethics committee of the University of Tübingen [29]. All studies were in accordance with the declaration of Helsinki (2013, 2016, 2024).

## Electrode localization and labelling

Electrode localization for P01-P08 and P10 (Fig 1) was performed by co-registering high resolution post-implantation 3D-Computerized Tomography (CT) scans (Philips Tomoscan SR7000, Best, The Netherlands; voxel size 0.49x0.49xT mm, with T ranging from 0.45 to 1 mm) with preoperative T1-weighed anatomical scans on a 3T Magnetic Resonance system (Philips 3T Achieva, Best, The Netherlands; isotropic voxels size of 1 mm). Followed by the detection of electrodes on the CT, computing each their center of mass, and then projecting them onto the individual freesurfer cortical surface while correcting for brain shift using local grid geometry [31,32].

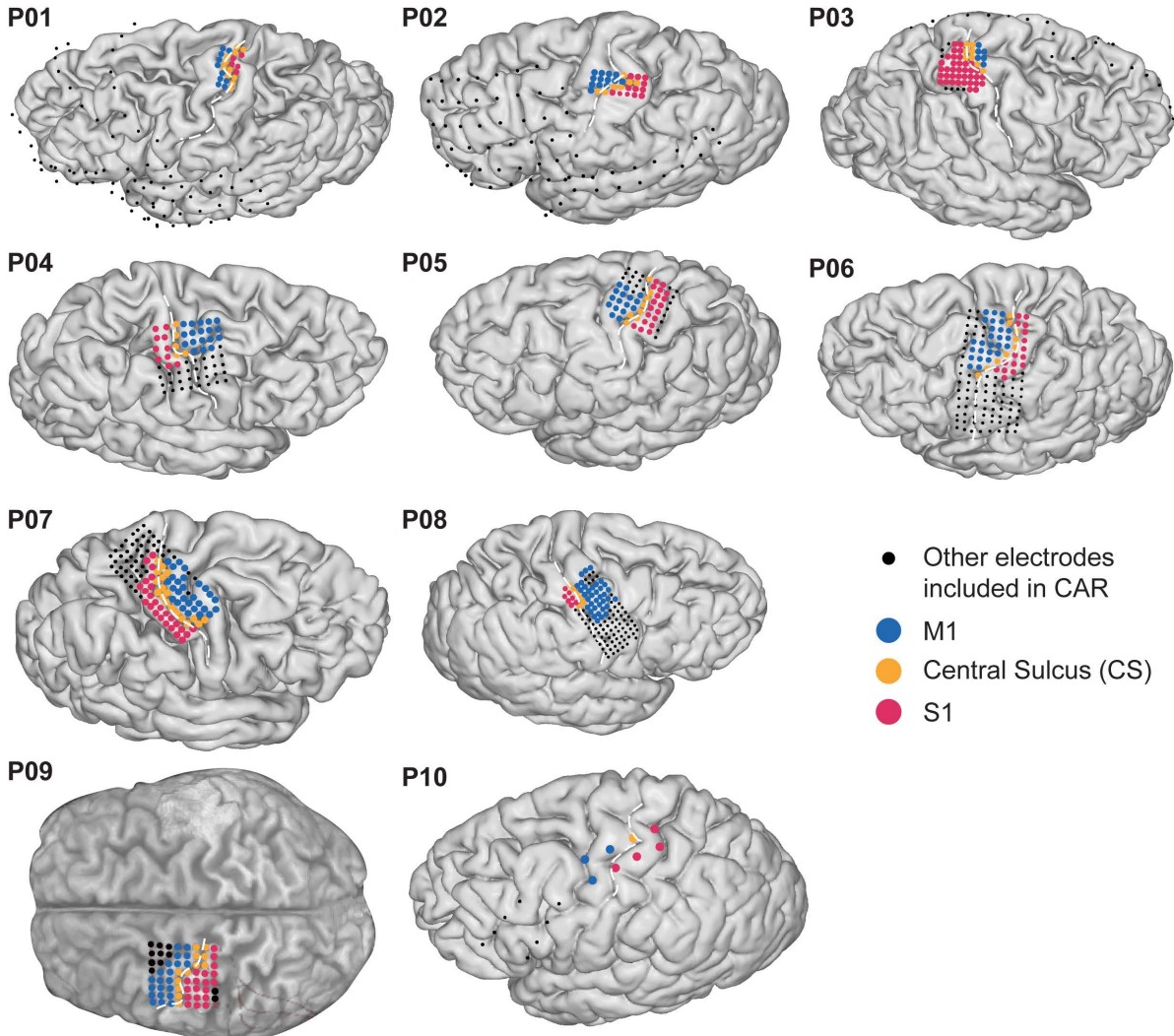

**Fig 1. ECoG grid localization.** Projection of electrodes onto brain surface renderings for all participants. Electrodes were labeled and colored as M1 (blue), S1 (pink) or central sulcus (CS; yellow) depending on their location over the sensorimotor cortex. All electrodes displayed were included in common-average reference (CAR), including the ones displayed in black. Central sulcus is highlighted with a dashed white line on the individual brain surface. Only electrodes displayed in the figure were included in the analysis. Electrode labeling of P09 was visually specified based on localization picture provided by Gharabaghi and colleagues [29].

For participants P01-P08 and P10, we identified electrodes that were strictly over the hand-knob region by constraining the coordinates of the electrodes that fall within a region expected to have a response to hand movement. The coordinate constraint was determined by mapping the subject electrode coordinates to a common space using a Cgrid method (https://github.com/mathijsraemaekers/Cgrid-toolbox) that converts individual surface sensorimotor spaces to a 2D standard cartesian grid of 40x80 tiles [33,34]. Then we identified a minimum and maximum y-coordinated ($41 \leq y \leq 65$) based on the functional Magnetic Resonance Imaging beta-maps of 20 healthy participants performing a hand localizer task reported in [33]. Subsequently, electrodes were visually labelled as 'M1' or 'S1' electrodes depending on their location (M1 or S1, respectively). The labelling was conservative in a sense that electrodes that could not be attributed with certainty

over M1 or S1, or were located over other regions, were labelled as either central sulcus ('CS') or 'Other'. Electrode labeling for P09 was visually estimated from electrode localization images provided by Gharabaghi and colleagues [29].

## Task and behavioral response

Participants were asked to perform an individual finger movement task with the hand contralateral to the implanted hemisphere. They were instructed to either move the thumb, index, or little finger. For P01, P02 and P04-P08, 30 movement trials per finger were conducted in randomized order. For P03 the task was adapted to include movement of all five fingers with 10 trials per finger, but only data for the three fingers (thumb, index and little) were included in the analysis. The intertrial duration was 4.4-5s and the participants were asked to flex the respective finger once (P04-P09) or 2–3 times (P01-P03). For P04-P08 the intertrial interval included an explicit rest period of 4s. P09 and P10 were instructed to attempt to move their (missing) fingers: P09 attempted to extend all fingers for 2s (24 trials per run, [29]), whereas P10 attempted continuous tapping of fingers against the thumb for 15s (10 trials, [30]). For P01-P08, movement was recorded using a dataglove (5DT Data Glove 5 Ultra; https://5dt.com/5dt-data-glove-ultra/). Please note that task paradigm and design for participants P1–P8 and P9–P10 were not identical, as the datasets were originally collected for other research purposes. Nevertheless, all participants were instructions to move their fingers.

## ECoG acquisition and data preprocessing

ECoG data was recorded using a 128 channel Micromed LTM system for P01-P03 and P10 (Treviso, Italy; 22 bits, hardware band-pass filter 0.15–134.4 Hz; sampling frequency 512 Hz), and Blackrock system for P04-P08 (Microsystems LLC, Salt Lake City, USA, digital band-pass filter 0.3–500 Hz; sampling frequency 2000Hz). Data from P09 was recorded with a monopolar amplifier (BrainProducts, Munich, Germany; high-ass filter 0.15 Hz; sampling frequency 1000 Hz; [29]). Preprocessing was performed in MATLAB® using the Fieldtrip toolbox [35]. In a first step, bad channels were identified (e.g., in case of broken leads or flat/noisy signals; [36]) and excluded from the rest of the analysis. We performed re-referencing using a common average reference (CAR) of all remaining channels. For P01-P03, clinical channels, recorded with the same amplifier and stored in the same file as the HD channels, were also included in the re-referencing. For remaining participants, clinical electrodes were recorded with a different amplifier and therefore not included in this analysis. In a next step, we applied a bandpass filter to equalize across amplifiers (0.15–134 Hz), as well as a 50 Hz notch filter. Data was subsequently divided into trials and bad trials identified by inspection of the dataglove and raw signal were excluded from further analysis (maximum 4 trials excluded per participant and finger).

## Trial alignment

Continuous ECoG signals of P01-P08 were divided into trials using movement-onset markers (MOM) extracted from the dataglove traces. MOMs were detected in a semi-automated fashion by convolving the dataglove with a derivative of a Gaussian to detect first deflection after the cue and then manually corrected by visual inspection.

For participants who attempted movement (P09 and P10) trials were aligned using the gamma-slope marker (GSM) method [37]. This method aligns trials based on the across (significant) channel high-frequency band (HFB) power response to the task. To this aim, first channels showing a significant task response were selected using an independent t-test (p < 0.05) comparing mean power of the pre-stimulus interval (P09 [−0.5, 0]s, P10 [−10, 0]s) and post-stimulus interval (P09 [0.5, 1]s, P10 [0,10]s) centered around the cue. Different trial lengths were chosen due to differences in task design, with trials for P09 being implemented in an event-related design and for P10 in a block design structure. In a second step, the mean HFB trace (60–100 Hz for P09 and 30–100 Hz for P10) was computed across all channels that show a significant response to the task. Third, for every trial the minimal horizontal distance between the mean HFB trace and a sliding slope line segment was computed and the corresponding time point was extracted as a marker for movement

onset. Trials were then aligned to the marker per trial, preserving any onset differences between electrodes. Trials where the GSM did not converge were excluded from further computations.

### Feature extraction

Spectral features were computed with a Morlet wavelet transform (width 7 and gwidth 3) using the ft_freqanalysis function in Fieldtrip [35]. Further analyses focused on power data: for P01-P09 alpha (8–12 Hz), beta (13–30 Hz), combined low-frequency band (LFB, 8–30 Hz), and high-frequency band (HFB, 60–100 Hz) were included. For P10 frequency bands were selected based on previous work with this participant [38] that showed only clear responses in other bands: alpha (8–12 Hz), beta (13–20 Hz), LFB (8–20 Hz) and HFB (30–100 Hz). Power was extracted per trial (-1s to 2s around the marker), channel and frequency bin (steps of 1 Hz), and averaged across frequency bins. We confirmed that the chosen frequency band ranges fit the participants by visually inspecting the time-frequency plots for the thumb movement for decreases (LFB) and increases (HFB) in power within those ranges (see S1 Text in Supporting Information for identification of frequency bands).

### Electrodes with significant response to the task

In order to identify neural onset per channel and frequency band, we applied constraints to the channels included for analyses. First, we determined which channels had a significant response to the task in each frequency band separately (alpha, beta, LFB and HFB). This could lead to different channels included in the analysis for different frequency bands. For each task and participant (P01-P08), significant channels were identified by means of a one-sided paired t-test (p < 0.05, no multiple comparison correction) between the mean pre-marker period (−1 to −0.5s) and a post-marker interval (0 to 0.5s) per trial. For P09 and P10 whose trials were aligned to GSM, significance in the HFB was established based on significance testing during GSM alignment and was not retested (see section Trial alignment). For alpha, beta and LFB, significant channels identified during GSM alignment were further evaluated for a significant response in the respective frequency band. To estimate the spatial extent of each frequency band and finger we computed the coefficient of determination (signed-$r^2$) between frequency band power and task design (active vs rest) per channel. Per frequency band, channels that showed significant increase (HFB) or decrease (alpha, beta and LFB) and that were inside the hand region (see section Electrode localization and labelling) were used for neural onset detection for that respective frequency band.

### Neural onset detection

Movement initiation was estimated by detecting the 'neural onset' of each frequency band separately. While for alpha, beta and LFB the neural onset corresponded to the onset of a decrease in power [39], for the HFB the neural onset corresponded to the onset of power increase [40]. Neural onsets were determined per channel, finger and frequency band using a multiple-step procedure:

- First, we smoothed all trials of each frequency band with a time window of 0.5s. Trials were epoched with a temporal window spanning from -1s to 2s around the marker (MOM or GSM).

- Second, we computed the mean over smoothed trials for each significant channel (see previous section), normalized this mean using a z-score function (centering the mean at 0 and scaling the signal's standard deviation to 1), and performed a baseline correction by subtracting the corresponding mean of the pre-movement baseline time-interval [−1, −0.5]s from each time-point of the signal of each individual channel.

- Third, the neural onset point of each channel was determined using a threshold (Fig 2A). Instead of choosing a threshold value manually, the threshold value was optimized per participant, finger and frequency band. The threshold

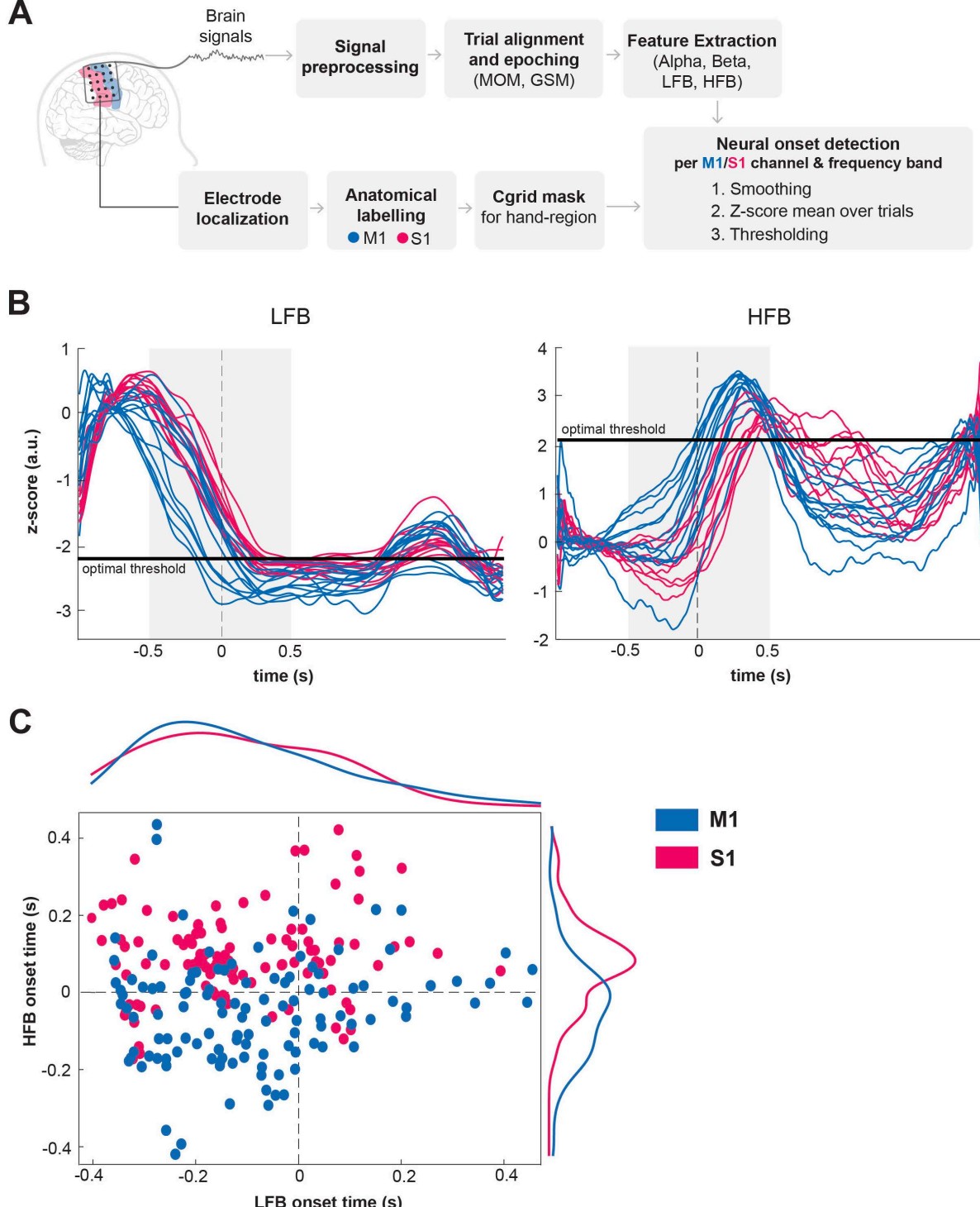

**Fig 2. Thresholding method for computation of neural onsets. A)** Overview of the processing steps towards the detection of neural onsets. GSM – Gamma-slope marker. **B)** Example of normalized mean LFB (left panel) and HFB (right panel) traces over trials per channel (z-score, arbitrary units, a.u.) for P02 and thumb. Blue curves indicate channels located over M1 and pink curves channels over S1. Threshold was optimized in the window −0.5 to 0.5s (gray shaded region) around movement onset marker (MOM; t = 0s; vertical dashed line). Optimal threshold is indicated with solid horizontal

black line for LFB and HFB separately. **C)** Scatter plot with marginal histograms of HFB and LFB onset points of P01-P08 for the thumb. The marginal histogram curves on top and on the right give an indication of the distribution of onsets for LFB and HFB, respectively.

optimization procedure was chosen to guarantee objective value selection while still accounting for interindividual neural response differences. The threshold value was optimized to describe the maximal response in the form of power increase (HFB) or decrease (alpha, beta and LFB) that was reached by a majority of channels within the time window −0.5s to 0.5s around movement onset. For the optimization we started with increasing/decreasing the threshold value starting at 0.4 (HFB) or −0.4 (alpha, beta and LFB) in steps of 0.1. The first goal of the optimization was reached when the threshold value was higher/lower than the mean power of all but two channels at time point −0.5s. Then, we kept increasing/decreasing the threshold value until the number of channels reaching the threshold started to drop. This extra step ensured that the threshold value was consistently chosen across participants, frequencies and fingers, thereby preserving the systematic difference between channels. As a consequence of the method the neural onsets were shifted to after movement onset. Nonetheless, as can be seen in Fig 2A, the real onset of activity in both HFB and LFB generally started before movement onset (t = 0s), which was also confirmed by extracting the neural onsets without this extra rule (see also S1 Fig in Supporting Information).

- Last, to identify timing patterns on a group level we displayed the LFB and HFB neural onsets per channel in a scatter plot with marginal histograms using a mean threshold across participants (Fig 2B).

### Statistical analysis

For each participant, the neural onset values were averaged separately for each finger and cortical location (M1 vs S1). The mean neural onsets of M1 and S1 channels were then statistically compared using a linear mixed-effects ANOVA in R [41] with cortical location and finger as within-subject factors and participant as the repeated-measures grouping factor. Post-hoc pairwise comparisons were performed using estimated marginals means and Tukey's correction was used to adjust for multiple comparisons. Analysis was performed per frequency band separately. Using this approach, we tested for differences in neural onsets between the two regions, by collapsing the spatial information while also accounting for differences across fingers. Post-hoc Welch two-sample t-test was used to identify participants with significant difference between M1 and S1 channels ($p < 0.05$, Bonferroni corrected for number of participants excluding participants with only 2 channels or less per cortical region). Additionally, to uncover the spatial mapping of movement initiation we also displayed neural onset on individual cortical surfaces and identified the region (M1 or S1) corresponding to the earliest (smallest) neural onset within one participant/finger. In addition, to summarize the cortical distribution of the earliest channel onsets across participants, we computed the percentage of M1 channels within the first (earliest) quartile (25%) of all M1/S1 channels, for each participant, finger, and frequency band.

### 3. Results

### Electrodes with significant response to the task

The spatial extent of each finger (or fingers) was mapped by means of signed-$r^2$ to the task, on all electrodes that showed a significant response to the task (Fig 3). On average for all 8 participants performing single finger movement, LFB showed the highest proportion of channels with significant response to the task (94%), followed by beta (91%), alpha (88%) and HFB (84%). While beta and LFB had the highest number of responsive channels (see also S1 Table), HFB exhibited a more distinct spatial map per finger (Fig 3B). Even though P01 and P02 have the smallest grids (32 channels), they capture HFB activity for each finger. Yet for other participants, such as P03, there is a stronger response for

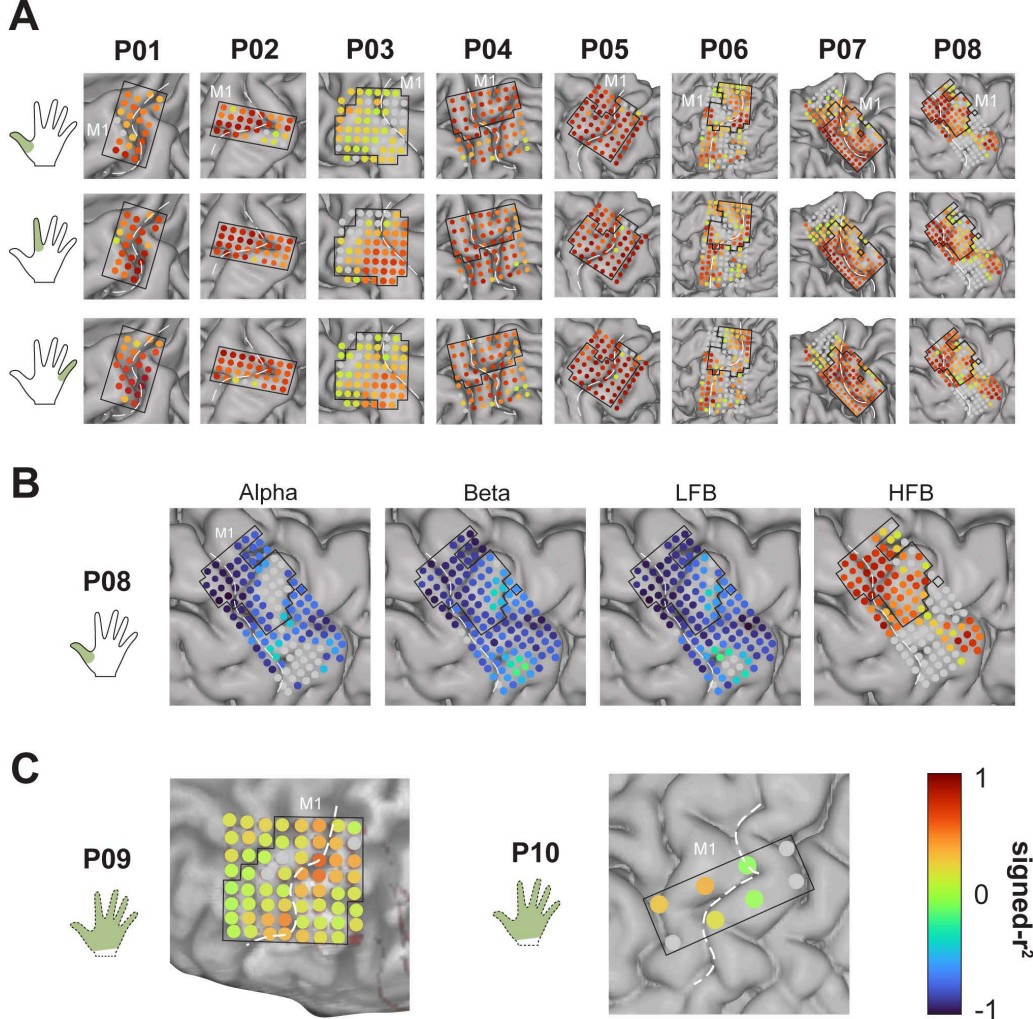

**Fig 3. Significant response to the task and signed-r² maps. A)** High-frequency band signed-$r^2$ values indicating a significant response to the task (active vs rest) for each participant and finger. **B)** Signed $r^2$ values for each frequency band (alpha, beta, LFB and HFB) for one exemplar participant (P08) and one finger (thumb). **C)** High-frequency band signed-$r^2$ values for attempted movement of fingers for P09 and P10. Dark blue values indicate a high negative correlation (signed-$r^2$ = −1) and dark red values indicate a high positive correlation to the step function (signed-$r^2$ = 1). Non-significant channels are indicated in gray. Central sulcus is indicated with white dashed line. Hand pictogram indicates which finger was moved. Primary motor cortex side is indicated by the label 'M1'. A-C) Electrodes delineated by black box were used for neural onset analysis.

only some fingers (for example, index finger compared with thumb). As expected, the participants with largest grids also display a more extensive topography of the three fingers.

In contrast, for attempted movements (P09 and P10) the average number of significant channels in HFB was two to three times larger than the lower-frequency counterparts (80% compared with 30–43%), albeit with smaller $r^2$ values (Fig 3C). Of note, for these two participants, only the channels used for GSM onset detection were tested for significance for the lower-frequency bands. Because of the lower number of included channels for P10 (in total 8) and the fact that only 5 were significant for HFB, the number of significant channels for the lower-frequency bands was limited to one for alpha and none for beta or LFB.

## Comparison of neural onsets between M1 and S1

To identify timing patterns, onsets for LFB and HFB were first displayed against each other on a scatter plot (Fig 2B) for all able-bodied participants (P01-P08) and thumb movement. We found that for the thumb HFB data, M1 neural onsets frequently occurred before the S1 neural onset, with the earliest activating channels being predominantly located in M1. For LFB there was no clear difference in onset. We verified the effect in HFB by directly comparing the onsets in S1 to those in M1 for each finger task (Fig 4). A mixed-effects ANOVA revealed a significant difference in mean onset times between M1 and S1 electrodes consistent across participants and fingers ($F_{(1,7)} = 17.77$, $p < 0.01$). Post-hoc tests confirm that onset time is significantly lower for M1 channels compared to S1 channels ($t(40) = -2.85$, $p < 0.01$). The average difference between M1 vs S1 onset was $-129ms \pm 80ms$ for thumb, $-91 \pm 104ms$ for index and $-105 \pm 66ms$ for little finger (S2 Table). There was also a significant difference between fingers ($F_{(2,14)} = 3.931$, $p < 0.05$) but no significant interaction effect between region (M1 and S1) and fingers ($F_{(2,14)} = 0.31$, n.s), suggesting

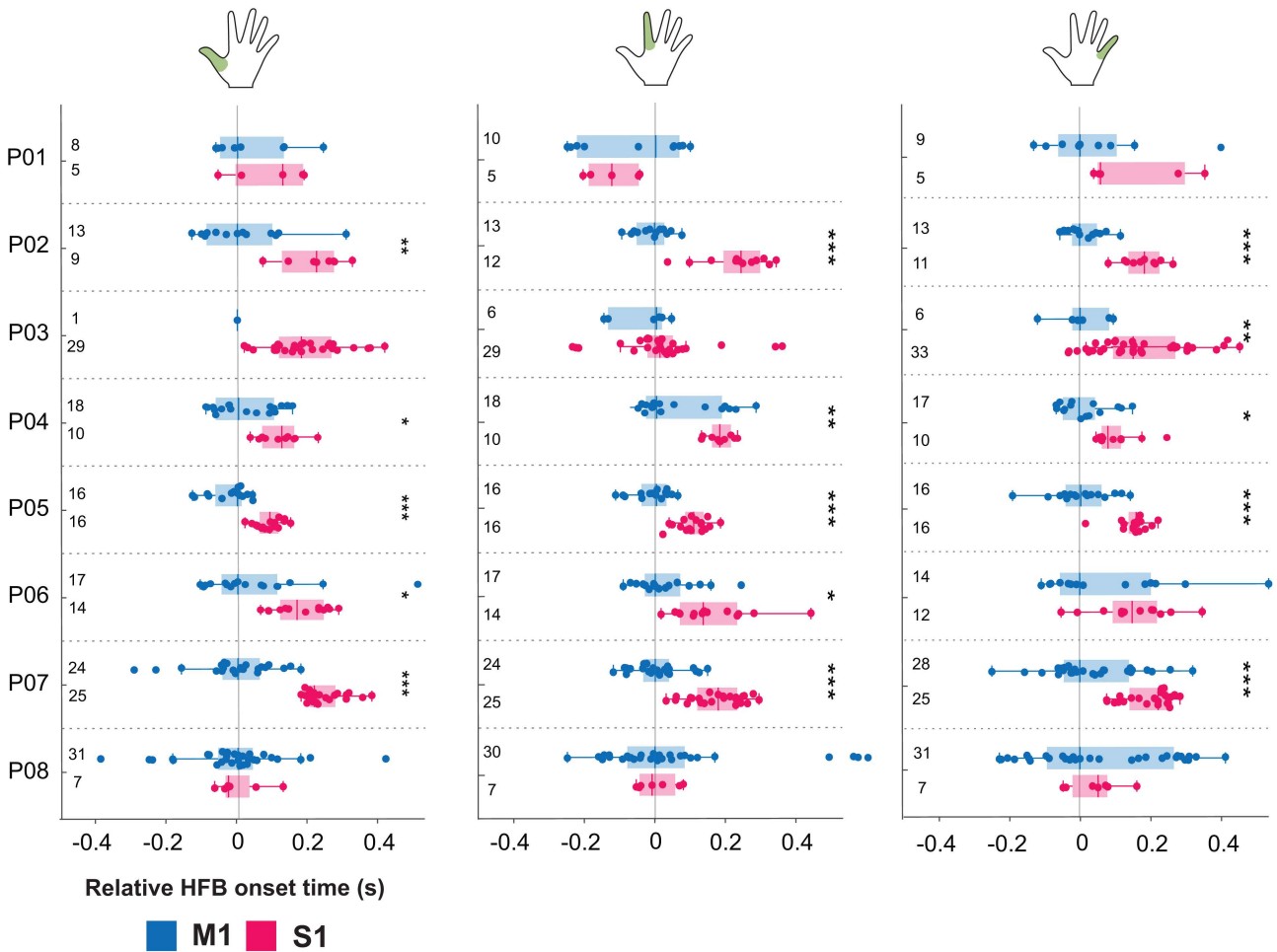

**Fig 4. Neural onsets during finger movement.** HFB neural onset time (in seconds) for M1 and S1 channels, for the thumb (left panel), index (middle panel) and little finger (right panel) for all abled-bodied participants (P01-P08). The number of included channels per cortical region are indicated below each boxplot. For reference, horizontal lines at t = 0s align the median onset time for all M1 channels for each participant. Vertical dashed lines separate participants. Hand pictograms indicates which finger was moved. Participants with a significant difference between M1 and S1 (post-hoc t-test) are indicated with * p < 0.05, ** p < 0.01 and *** p < 0.001. P-values were Bonferroni corrected per finger for the number of participants excluding those with 2 or less electrodes in M1 or S1 (n = 7 for thumb and n = 8 for index and little finger).

that the M1-S1 relative onset effect is consistent across fingers. When looking at the spatial map of the neural onset (Fig 5), we found that the first activating channels (white asterisks) were always (except for index finger of P03, see Fig 4B) located in M1, which is strengthened by the high percentage of M1 channels in the set of channels in the earliest onset quartile: on average 96% for thumb, 89% for index and 94% for little finger (S2 Table). We further compared the HFB neural onsets with those detected using more permissive thresholds (see section Neural onset detection) resulting in earlier onset times. As expected, the majority of neural onsets occurred before movement onset (S1 Fig) while differences between M1 and S1 were preserved.

In contrast, we did not find a significant effect in LFB when comparing M1 and S1 onsets across participants ($F(1,7) = 0.91$, n.s.; S2A Fig). However, there was a difference between fingers ($F(2,14) = 5.37$, $p < 0.05$) but no interaction between finger and cortical region ($F(2,14) = 0.52$, ns). It can be observed in S2A Fig that the onset difference between M1 and S1 in LFB differs considerably between participants. To investigate whether the LFB diffused pattern could be accounted for by the large frequency range (8–30 Hz), we tested this effect for the alpha and beta band separately (S2B, S2C Fig). Again, we found no significant difference of onset times between cortical regions for alpha ($F(1,7) = 0.73$, n.s.) nor for beta: $F(1,7) = 0.57$, ns). Between fingers there was no significant for alpha ($F(2,14) = 2.05$, n.s.) nor beta ($F(2,14) = 2.79$, n.s.). Further, there was no interaction effect for alpha ($F(2,14) = 1.23$, n.s.) or beta ($F(2,14) = 0.76$, n.s.).

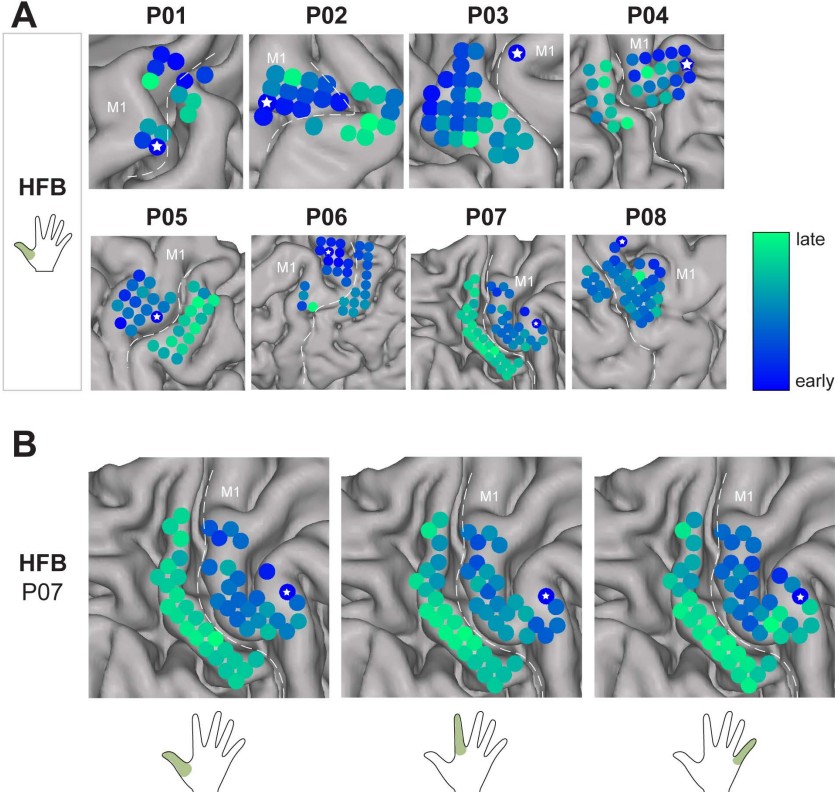

**Fig 5. High-frequency band neural onset time across the sensorimotor cortex.** (A) HFB neural onsets for thumb finger normalized per able-bodied participant (P01-P08). (B) HFB onset map for a representative participant (P07) for the thumb (left) index (middle) and little (right) fingers. (A-B) Earliest neural onsets are displayed in blue and latest onsets in green. The central sulcus is indicated with dashed white line and the side of the primary motor cortex is labelled with an 'M1'. White asterisks represents the earliest activating channel.

## Executed and attempted movement

Having established that HFB activity in M1 precedes that in S1 during executed finger movement, we evaluated whether this was also the case for minimal or no reafference during attempted movement (Fig 6). For two paralyzed participants P09 and P10 we aligned HFB neural responses using the GSM method. P09 has 19 electrodes over M1 and 18 electrodes over S1, of which 19 and 15, respectively, showed a significant HFB response to the task. P10 had 3 electrodes covering M1 and 4 covering S1, of which 2 channels over M1 and one over S1 showed a significant HFB response to the task. Overall, the distribution of neural onset of participants P09 and P10 shows a similar pattern as in able-bodied participants, with an earlier HFB onset in M1 channels (P09: mean $0.13 \pm 0.03$s; P10: mean $0.22 \pm 0.05$) compared to S1 channels (P09: mean $0.19 \pm 0.07$s; P10: 0.23, only one channel). In both participants the first activating channel was located in M1, and for P09 78% of M1 channels fell in the earliest quartile.

## 4. Discussion

The exact role of primary somatosensory cortex (S1) during movement initiation, execution and attempt is still elusive. In this study we quantified the timing dynamics between primary motor cortex (M1) and S1 in the presence and absence of movement and for different frequency bands. Our results show a clear activation pattern for high-frequency band (HFB) signals, where M1 channels activate before S1 channels on a participant- and group-level, but no consistent activation pattern in the low-frequency band (LFB). The directionality of information agree with the findings from HFB signals of non-human primates [17] and single unit spiking activity in humans [42]. Umeda and colleagues showed that although S1 becomes active before movement onset, increases in M1 activity consistently precede those in S1. Their results support the idea that S1 activity during voluntary movement can be decoded from M1 prior the arrival of proprioceptive and sensory inputs. In addition, Jafari and colleagues demonstrated that S1 single units in an individual with tetraplegia encode imagined reaching movements and may contribute to the motor engagement even in the absence of sensation. These results stand in direct contrast to the findings of Sun and colleagues, who reported earlier activation in S1 than in M1 [16].

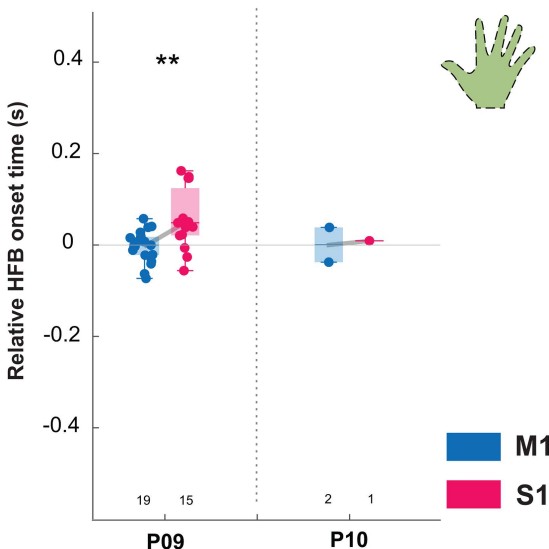

**Fig 6. Neural onset during attempted movement.** Relative HFB neural onset (in seconds, s) for attempted movement of the hand (P09 and P10). Vertical dashed line separates participants. M1 channels are indicated in blue, while S1 channels are indicated in pink. The number of included channels per participant are indicated below each boxplot. Horizontal gray line ($t = 0$s) indicates the median onset time for M1 channels. Significant difference between M1 and S1 for P09 (post-hoc t-test) is indicated with ** $p < 0.01$.

However, this study compared the neural onset between limited number of sparsely placed electrodes (spaced at least 1 cm apart and often spreading beyond the hand region), which could lead to a biased interpretation of timing given that one can certainly find one electrode in S1 that activates before one electrode in M1. In contrast, our analysis uses closely placed electrodes (3–4 mm apart) within the hand area of M1 and S1, providing a much denser spatial coverage. Interestingly, another study by Witham and colleagues did find the same direction of information (M1 before S1) but instead in the beta band [15]. They report a directed phase coherence (Granger causality) from M1 to S1 using intracortical local field potential recordings, which are much more detailed than ECoG recordings. Yet, given the inter-subject variability we found in our study, it remains unclear whether this activation pattern would also sustain in a larger sample size.

### 4.1 Different roles of low- and high-frequency features

The spectral content of neural signals has classically been divided into several rhythms or frequency bands, each typically presenting different neural processes [43]. While LFB oscillations are visible from the synchronized increased in excitability of thalamo-cortical systems [44–46], HFB signals are more often associated with activity of neurons and dendritic potentials [40,47–50].

Here we show that HFB signals in M1 precede those in S1, both in the presence and absence of movement. In contrast, LFB signals show no consistent tendency across subjects or fingers in able-bodied participants. This difference could be driven by the tight relationship between HFB and movement. Indeed, HFB is a relative broad frequency band above 40 Hz that is recorded in the contralateral hemisphere of hand moved that appears to be tightly linked to the initiation of a motor response, closer than alpha, beta or low gamma activity [40,47,51]. This feature seems to directly reflect movement performance (e.g., [52]), to be equally found in M1 and S1 (e.g., [37]), and to be present in the absence of movement (e.g., [21,29,30]).

One could argue that the lack of consistency in the LFB could be driven by the dilution of two rhythms in one band, more specifically the alpha and beta rhythms. While the event-related-desynchronisation (ERD) in the alpha and beta band is associated with an increased excitability of the respective cortical brain region [39], they differ in location and assumed function. Alpha ERD has been found to be centered above the postcentral gyrus (S1), associated with somatosensory functions, while beta ERD has been assumed to have its neural origin in the precentral gyrus (M1) and is consequently associated with motor functions [53,54]. In line with this theory, Ohara and colleagues found the highest ERD dampening in S1 in the frequency range 10–15 Hz and in M1 in the frequency range around 20 Hz [55]. Interestingly, when looking at the activation order between M1 and S1 the authors also report inconsistencies across participants and alpha-beta bands. This is in line with our varying results after splitting LFB in alpha and beta bands. This poses the question whether activation patterns in the lower frequency bands are meaningful as representations of travelling activation between M1 and S1 or whether they represent distinct functions in M1 and S1 completely, while the communication between M1 and S1 cortices is driven by higher frequency bands. In order to fully understand the temporal signature of LFB, one could argue that the frequency range of LFB decomposition should be determined on a participant level using for example irregular-resampling auto-spectral analysis method, thus accounting for interindividual differences [56]. Another source of potential cofounds could be the fact that our able-bodied participants suffered from intractable epilepsy, which can be associated cortical lesions that may affect the neural signals. Yet, we found that was only the case for 2 out of 8 participants (P05 and P07), thus not accounting for all the variability in the beta band.

### 4.2 Sensorimotor controls models

The interaction between M1 and S1 can also be interpreted in light of motor control theories, which aim to model and explain the complex flow of information between cortical (and subcortical) regions [57]. Early control models first attempted to describe the adaptation of movement through mathematical formulas and engineering approaches [3]. While the optimal feedback control model (OFC), an adapted variant of earlier control models, has been established for

describing movement initiation in the brain, the active inference (AIN) model has also recently emerged as an alternative model [58]. The AIN and the OFC model differ in both theoretical as well as biological aspects. One point of difference between these two models is how the sensorimotor cortex integrates movement. While in the OFC model there is a clear separation between the motor and the somatosensory system, with the latter only being relevant for providing feedback, in the AIN model these two regions are more intertwined and are responsible for the prediction of sensory consequences. There are some commonalities between both models, especially regarding the sensory input to M1 to update motor output. This input from S1 to M1 may represent different phenomena (i.e., state estimates in OFC vs prediction errors in AIN). In OFC a forward (driving) model uses an efference copy from M1 to generate a prediction of the proprioceptive outcome sent to S1. In contrast, in AIN a generative model uses S1 to create proprioceptive predictions which are sent to M1. Our results support the concept of information being flown from M1 to S1. However, whether this information is a result of an efference copy in OFC or somatosensory prediction in AIN remains elusive. Interestingly, Borich and colleagues suggested that both models could coexist, if the recruitment of one or other would depend on the complexity of the movement (e.g., simple finger movement vs complex gestures), where OFC would primarily control simple movement. This hypothesis would suggest that our abled-bodied participants (who moved individual fingers) likely obey an OFC framework.

### 4.3  Interpreting the neural onset with respect to movement onset

In this study we focused on detecting the timing difference between M1 and S1 signals, by applying the same method to all electrodes. Several methods for alignment of the data can be used, such as movement onset as detected with dataglove [16] or muscle onset detected with EMG [15,17]. These two methods offer a far better alignment compared with cues, given that they correct for reaction time. In this study we used MOM to align the data of able-bodied participants, since they performed relatively simple tasks that can easily detected using a dataglove. However, to verify that MOM were not introducing a bias in our methods we compared the results using EMG onset markers in one participant who had an EMG sensor placed over the thumb. Our results confirm that MOM are an appropriate method to align the data and to interpret neural onset (see S2 Text in Supporting Information). Of note, the neural onsets detected with EMGM seem to be later than the ones with MOM, which can be explained by the surface EMG electrode was sub-optimally placed. When no movement was performed we aligned the data instead with respect to the increase HFB (gamma-slope marker, GSM), a method previously shown to preserve relative timing between electrodes [37].

Given the nature of the neural onset detection method used, the exact time with respect to movement onset (MOM, EMGM or GSM) may not represent the actual neurophysiological onset because of the averaging, smoothing and thresholding applied. We confirmed that movement onset occurred in general after M1 and S1 by comparing movement and HFB timing, showing that in general neural onsets occur before movement onset and that the neural onset detection with our method does not depend on the way of thresholding. Our results show the majority of early onset channels of M1 and S1 activating before MOM, which is within the range of observed delay between neuronal firing and muscle activity [59,60]. Ideally, neural onsets would be detected on a single trial basis per channel, yet the noisy nature of spectral features, particularly HFB, often require the averaging across trials for detection (e.g., [17]). Modelling of the spectral signal using a mixture of Gaussians may present a robust way to detect onset on a single-trial for the HFB, but a method that would work across multiple spectral and temporal features, specifically with complex shapes like motor related cortical potentials, remains to be devised.

### 4.4  Strengths and limitations

One of the strengths of our study is the large ECoG dataset of 10 participants, of which two performed attempted movement and had minimal or no reafference. In particular, we take advantage of exclusive high-density ECoG grids with an interelectrode distance of 3–5 mm in 9 out of 10 participants, providing us with a unique coverage of the hand-knob area in the sensorimotor cortex.

A limitation of our study is the varying grid placement across participants, at times sub-optimal to detect specific fingers. For example, P03 shows barely any response to thumb, compared with index and little fingers (Fig 3A). While the sensorimotor cortex is known for its tight topographical representation of the hand [6], the exact center of each finger can vary across M1 and S1 [61,62]. Even though a smaller coverage of the hand-knob may limit the detection of a (true) neural onset, larger grids with more than 64 electrodes grant a full map of the hand-knob, hence increasing the chance of detecting the first active cortical patch in M1 and S1 during finger movements. In addition, ECoG grids can only map the surface of the cortex (gyri), being understandably blind to approximately 2/3 of the cerebral cortex embedded in the depth of the sulci [63]. Even though other non-invasive technique, such as magnetic resonance imaging, allow for a full cortical map, these lack the temporal resolution to study neural temporal dynamics. Here we intentionally avoided electrodes over the central sulcus since they do not clearly represent M1 or S1 but the net activity from both. Only a study combining surface ECoG recording with intracortical microelectrode and depth stereo-encephalography (sEEG) recording on a large number of participants will allow to fully map the temporal progression of neural signals in sensorimotor cortex during movement initiation.

Another limitation is that task parameters differed across participants (e.g., fingers tested, specific instructions, and inter-trial timing). These differences were accounted for in the analysis to ensure comparability: all participants were instructed to move one or multiple fingers during active trials, which reliably activates the hand region of the sensorimotor cortex (e.g., [5,37,61]). For alignment, we used movement onset for able-bodied participants and the GSM for participants P9–P10, ensuring all trials were referenced to a common point for comparison across channels and participants.

Last, in this study we examined the timing dynamics during the performance of actual movement in 8 participants. This inherently leads to the overlap of top-down motor commands and bottom-up sensory input and makes it impossible to draw causal conclusions of the origin of activation, especially pre-movement activations in S1. We addressed this by comparing our results in two participants with minimal or no reafference (P09 and P10). Our preliminary findings show a similar pattern in HFB signals to that observed in able-bodied participants. However, these observations need to be validated in a larger and more diverse cohort, including individuals with different neurological disorders. Indeed, several conditions, such as ALS, can lead to reduced grey-matter thickness [64,65] and decreased HFB signals over M1 [66], which may alter these dynamics.

Looking ahead, emerging approaches combining ECoG with optogenetics offer a promising avenue for disentangling the respective contributions of M1 and S1. Such methods allow transient, region-specific suppression or activation [67,68], enabling causal tests of how perturbing one area influences movement production and the neural responses of the other. Incorporating these techniques in future studies could provide a powerful way to isolate top-down and bottom-up components of movement-related activity.

## 5. Conclusion

The exact mechanism behind movement control in the sensorimotor cortex remains elusive. Understanding the timing and direction of information between M1 and S1 can help further unveil sensorimotor integration. We used high-density ECoG recording to determine the timing difference between M1 and S1. We showed that high-frequency band signals of M1 activate before S1 during the movement of individual fingers even in the absence of reafference. In contrast, low-frequency signals showed no consistent onset preference for M1 or S1. These results call for an extended role of S1 during movement production beyond processing sensory input. Future work should further promote the full comprehension of M1 and S1 integration, as it is key for better understanding neuromuscular disorders and for the optimal development of neurotechnology.

## Supporting information

**S1 Text. Identification of frequency bands.**
(DOCX)

**S1 Table. Channels included in the analysis.**
(DOCX)

**S2 Table. Timing difference between M1 and S1.**
(DOCX)

**S1 Fig. Interpretation of neural onsets during execution of three fingers.**
(DOCX)

**S2 Fig. Neural onsets during execution for low-frequency bands.**
(DOCX)

**S2 Text. Movement onset detection based on muscle activity.**
(DOCX)

## Acknowledgments

The authors thank Dr. Alireza Gharabaghi and Dr. Georgio Naros for kindly sharing their data described in [29] (here P09), and for providing the relevant information to interpret the data.

## Author contributions

**Conceptualization:** Zachary V. Freudenburg, Mariana P. Branco.

**Data curation:** Sophia Gimple.

**Formal analysis:** Sophia Gimple.

**Funding acquisition:** Nick F. Ramsey, Mariana P. Branco.

**Methodology:** Sophia Gimple, Zachary V. Freudenburg, Mariana P. Branco.

**Resources:** Nick F. Ramsey.

**Supervision:** Zachary V. Freudenburg, Mariana P. Branco.

**Visualization:** Mariana P. Branco.

**Writing – original draft:** Sophia Gimple.

**Writing – review & editing:** Zachary V. Freudenburg, Nick F. Ramsey, Mariana P. Branco.

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
