## [Decision Letter · Decision Letter 0]

9 Dec 2025

PONE-D-25-59260Temporal responses in sensorimotor cortex during attempted and executed movementsPLOS One

Dear Dr. Branco,

Thank you for submitting your manuscript to PLOS ONE. After careful consideration, we feel that it has merit but does not fully meet PLOS ONE’s publication criteria as it currently stands. Therefore, we invite you to submit a revised version of the manuscript that addresses the points raised during the review process.

We look forward to receiving your revised manuscript.

Kind regards,

Onder Aydemir

Academic Editor

PLOS One

Journal Requirements:

“This research was funded by the Dutch Research Council (NWO): NeuroCIMT project (grant 14906, NFR) and PANDA project (grant 19072, MPB).”

“This research was funded by the Dutch Research Council (NWO): NeuroCIMT project (grant 14906, NFR) and PANDA project (grant 19072, MPB).”

“This research was funded by the Dutch Research Council (NWO): NeuroCIMT project (grant 14906, NFR) and PANDA project (grant 19072, MPB).”

6. In the online submission form you indicate that your data is not available for proprietary reasons and have provided a contact point for accessing this data. Please note that your current contact point is a co-author on this manuscript. According to our Data Policy, the contact point must not be an author on the manuscript and must be an institutional contact, ideally not an individual. Please revise your data statement to a non-author institutional point of contact, such as a data access or ethics committee, and send this to us via return email. Please also include contact information for the third party organization, and please include the full citation of where the data can be found.

Additional Editor Comments:

Dear Authors,

Thank you for submitting your manuscript. Both reviewers find the study scientifically valuable, with strong potential impact, particularly regarding the consistent finding that M1 HFB activity precedes S1 during movement. However, they also identify several substantive issues that must be addressed before the manuscript can be considered for further evaluation. On the basis of their comments and my own assessment, a major revision is required.

Please address the following points in your revised submission

Reviewers' comments:

Reviewer's Responses to Questions

**Comments to the Author**

1. Is the manuscript technically sound, and do the data support the conclusions?

Reviewer #1: Partly

Reviewer #2: Yes

2. Has the statistical analysis been performed appropriately and rigorously? 

Reviewer #1: Yes

Reviewer #2: Yes

3. Have the authors made all data underlying the findings in their manuscript fully available?

Reviewer #1: No

Reviewer #2: No

4. Is the manuscript presented in an intelligible fashion and written in standard English?

Reviewer #1: Yes

Reviewer #2: Yes

5. Review Comments to the Author

Reviewer #1: This technically solid and important study investigates the temporal relationship between primary somatosensory cortex (S1) and primary motor cortex (M1) during both executed and attempted movements to reassess whether S1’s role extends beyond purely reafferent sensory processing. The authors analyze high-density ECoG recordings from eight participants with grids over the hand area of sensorimotor cortex (execution cohort), comparing the timing of responses in electrodes labeled M1 versus S1 across high-frequency band (HFB) and low-frequency band (LFB) activity, and then “verify” the timing pattern in two additional participants instructed to attempt movements with minimal sensory feedback. Their main finding is that HFB activity in M1 precedes S1 consistently across participants, even during attempted movement. LFB activity shows no consistent ordering. The authors interpret this as evidence that S1 is engaged around movement initiation, not merely a passive recipient of feedback.

Strengths

ECoG affords excellent temporal resolution and spatial coverage over peri-Rolandic cortex. The explicit contrast between executed and attempted movements targets the presence versus minimal presence of reafferent feedback. The band-specific analysis leverages the known link between HFB and local population spiking. If robustly quantified, a consistent M1-before-S1 lead in HFB, especially when overt feedback is minimized, does support an expanded view of S1 participation beyond classical sensory reception.

Weaknesses

However, several weaknesses limit the force of the conclusions as stated. First, the attempted-movement verification is based on only two participants, which might be underpowered for generalization. Second, concluding that S1 is “essential” for initiation overreaches the evidence: latency differences imply temporal engagement, not causal necessity (no perturbation, stimulation, or lesion-based tests are presented).

Recommendations for the authors

The movement-related HFB results in able-bodied participants are rigorous and convincingly show earlier onsets in M1 than S1. However, the central claim, an extended role of S1 during movement production beyond sensory input, rests primarily on the attempted-movement dataset, which includes only two participants and, for at least one of them (P10), sparse M1/S1 coverage with very few (2/1) responsive channels only for HFB activity and no channels for beta and LFB activity. Given this limited sample and coverage, the evidence for feedback-independent S1 engagement is suggestive but not robust. As it is written, the conclusion overreaches what the attempted-movement data can currently support. To strengthen the attempted-movement evidence, if possible, I recommend including data from at least one additional attempted-movement participant. If additional data are not feasible, I suggest tempering the claim to directional/indicative language rather than a strong verification of a feedback-independent S1 role.

I suggest that the hypotheses paragraph can be improved by, for example, (i) defining what you mean by “neural activity features” and “timing” (e.g., onset vs peak, how derived from ECoG) and briefly mention how dataglove-detected movement onset is aligned; (ii) specifying the frequency ranges for HFB and LFB and why both are analyzed (e.g., HFB: local population firing; LFB(alpha, beta): preparatory and feedback-related dynamics?); (iii) avoiding the claim of “no feedback to S1” in attempted movement and instead stating how minimal reafference is verified; (iv) making clear you predictions and hypotheses.

(Results Section 3.4: Executed and attempted movement): I suggest expanding this subsection, so the study’s central claim is quantitatively supported. For example, report subject-level statistics (e.g., latency differences) and electrode coverage and responsive-channel counts to contextualize P09 vs P10.

(Discussion Section, para 1): After the sentence noting that Sun et al., 2015 compared “pairs of (sparsely placed) electrodes,” please add one explicit line that states your analysis unit and coverage and contrasts this with their pairwise, sparse sampling. This concrete clarification will make clear why methodological choices can flip apparent M1-S1 ordering.

The Discussion reads coherently and supports the main results. To make it even stronger before acceptance, I suggest light-touch refinements: avoid causal/“essential” phrasing; keep the attempted-movement inference explicitly preliminary, refresh a few citations with recent studies (last 5–7 years), and tighten band-specific messaging by presenting LFB results as inconclusive here while keeping HFB central. In the limitations paragraph, add one or two actionable next steps to underscore feasibility.

Abstract (clarity and focus): Please streamline the abstract to center on the core, well-supported result: in HFB, M1 responses precede S1 across participants, with a preliminary observation that this pattern is also seen during attempted movement (n=2). I recommend omitting the LFB results from the abstract (or reducing to a brief “no consistent LFB effect”) since they are not shown in the main figures. Also, avoid causal language (e.g., “essential”). Briefly specify sample sizes (8 executed; 2 attempted) and what “timing” refers to (e.g., onset latency in HFB), and (minor) replace “prior or after” with “prior to or after.”

I would consider moving the full Methods section to the end (keeping a brief Methods Summary in the main text) and restructuring as Introduction → Results → Discussion → Methods. This format improves readability and mirrors the structure used by many journals. This would make the narrative flow cleaner while keeping all technical information accessible.

Minor recommendations for the authors

(Section 1: Introduction): Consider rephrasing the following part to calibrate the strength of claims about S1 (e.g., implying it can drive movement independently of M1) and to resolve minor grammar issues: …with S1 being able to drive movements independently from M1 (Matyas et al., 2010), as well as being directly linked to motor learning (e.g., de Freitas Zanona et al., 2022) and movement planning (Gale et al., 2021) through cortical feedforward mechanisms that predict and integrates a efference copies of motor commands with incoming sensory signals (Arikan et al., 2021).

(Section 2.3: Task and behavioral response): Please briefly explain the rationale for the differing task parameters across participants (e.g., fingers tested, instructions, inter-trial timing) and indicate how these differences were accounted for in the analysis to ensure comparability.

Revise main figure (Figure 4) for completeness. I suggest (i) including the attempted-movement participants (P09, P10) alongside P01–P08, (ii) reformat the plot with time on the x-axis and participants on the y-axis, and (iii) present three plots (A: thumb, B: index, C: little) with identical scales. This will make the M1→S1 timing pattern and its consistency across executed vs. attempted conditions immediately interpretable.

(Section 4.1: Discussion): In the sentence “Here we show that HFB signals travel consistently from M1 to S1… whereas LFB… seems to be subject-specific,” please avoid causal wording, “travel” implies directed flow, and use timing language instead (e.g., “M1 HFB precedes S1”). In the following sentences on alpha/beta, state explicitly that the LFB results are inconclusive in your dataset, and maybe move the participant-specific band-definition suggestion to Limitations/Future work rather than presenting it as an explanation of the current results.

(Section 4.4: Strengths & limitations): In the sentence “...two performed attempted movement and had therefore no sensory input,” please replace this with “minimal reafference.” If verification was performed, add a brief clause in the same paragraph indicating how (or whether) this was verified (e.g., EMG/kinematics, visual occlusion). As a general point, I recommend using “minimal reafference” rather than “no sensory input” throughout, as it better reflects the likely presence of unmeasured feedback.

Reviewer #2: This manuscript presents an important and ambitious investigation of the temporal dynamics between the primary motor cortex (M1) and primary somatosensory cortex (S1) using high-density ECoG recordings during both executed and attempted finger movements. The study leverages a rare and valuable dataset, including two participants without sensory input, and addresses a fundamental question in sensorimotor neuroscience with clear relevance for understanding motor control, neuromuscular disorders, and neurotechnology development. The results particularly the consistent finding that high-frequency band (HFB) activity in M1 precedes that in S1 are intriguing and potentially impactful. However, several essential aspects of the Methods require clearer description and stronger justification before the manuscript can be considered robust and reproducible. The major comments are outlined below.

1. Although the electrode localization procedure has been described in previous publications, the current manuscript should still provide a brief explanation of the algorithm and specify which software tools or custom code were used. Even if the method follows prior work, readers need enough methodological detail to understand and reproduce the electrode coregistration workflow. A short description of the steps and software would greatly improve clarity.

2. It would greatly improve clarity if the authors could include a methodological flowchart summarizing the full analysis pipeline. A visual overview of the key steps—such as electrode localization, channel selection, trial alignment, feature extraction, and onset detection—would help readers better understand the entire procedure and how each component fits together.

3. It is not clearly described how the MRI beta maps were obtained. The authors should indicate what task paradigm was used, which software package processed the fMRI data, and which MRI sequences were acquired. Important acquisition parameters for both T1-weighted structural imaging and fMRI should be included. In addition, clarification is needed regarding how and when the fMRI data were collected prior to ECoG implantation.

4. The manuscript does not provide sufficient information about the CT acquisition parameters. Key CT details such as slice thickness, voxel dimensions, scanner settings, and reconstruction methods should be reported, as these directly affect the accuracy of electrode localization.

5. The rationale for defining the high-frequency band (HFB) as 60–100 Hz is not clearly justified. It would be important to state whether other commonly used high-gamma ranges (e.g., 60–150 Hz or 60–200 Hz) were tested. Furthermore, the manuscript should explain why HFB is defined as 60–100 Hz for P09 but 30–100 Hz for P10. Because the conclusions rely heavily on HFB timing, a clearer justification or data-driven selection of frequency bands is necessary.

6. The procedure for performing the one-sided paired t-tests on electrodes showing significant task responses is not sufficiently explained. It is unclear whether the t-tests were conducted per channel, per participant, or across channels. The manuscript should specify the p-value threshold used for significance, and if multiple t-tests were performed, it is important to clarify whether any correction for multiple comparisons (e.g., Bonferroni correction or FDR) was applied. Without this information, it is difficult to assess the statistical rigor of the significance testing.

6. PLOS authors have the option to publish the peer review history of their article (what does this mean?). If published, this will include your full peer review and any attached files.

Reviewer #1: **Yes:**Olgerta Asko

Reviewer #2: No

---

## [Author Response · Author response to Decision Letter 1]

9 Mar 2026

We thank the editor and reviewers for their positive comments and suggestions. We address each comment separately in the Response to Reviewers document.

---

## [Decision Letter · Decision Letter 1]

6 Apr 2026

Temporal responses in sensorimotor cortex during hand movements

PONE-D-25-59260R1

Dear Dr. Branco,

We’re pleased to inform you that your manuscript has been judged scientifically suitable for publication and will be formally accepted for publication once it meets all outstanding technical requirements.

Kind regards,

Onder Aydemir

Academic Editor

PLOS One

Additional Editor Comments (optional):

Dear Author,

We are pleased to inform you that your manuscript entitled “Temporal responses in sensorimotor cortex during hand movements” has been accepted for publication. Both reviewers have carefully evaluated your revised submission and agree that you have satisfactorily addressed all previously raised comments and concerns. They find the manuscript to be significantly improved in clarity and strength, with interpretations that are now well supported by the data. Congratulations on this achievement, and thank you for your contribution to the field.

Sincerely,

Dr. Onder Aydemir

Reviewers' comments:

Reviewer's Responses to Questions

**Comments to the Author**

1. If the authors have adequately addressed your comments raised in a previous round of review and you feel that this manuscript is now acceptable for publication, you may indicate that here to bypass the “Comments to the Author” section, enter your conflict of interest statement in the “Confidential to Editor” section, and submit your "Accept" recommendation.

Reviewer #1: All comments have been addressed

Reviewer #2: All comments have been addressed

2. Is the manuscript technically sound, and do the data support the conclusions?

Reviewer #1: Yes

Reviewer #2: Yes

3. Has the statistical analysis been performed appropriately and rigorously? 

Reviewer #1: Yes

Reviewer #2: Yes

4. Have the authors made all data underlying the findings in their manuscript fully available?

Reviewer #1: No

Reviewer #2: No

5. Is the manuscript presented in an intelligible fashion and written in standard English?

Reviewer #1: Yes

Reviewer #2: Yes

6. Review Comments to the Author

Reviewer #1: Thank you for the careful revisions, which fully address my comments. The revised manuscript is clearer, stronger, and the interpretations are now appropriately supported by the data. I find the study technically sound, the analyses appropriate, and the manuscript suitable for publication. This study makes an important contribution by clarifying the temporal relationship between M1 and S1 during both executed and attempted movements, with implications for how S1’s role in movement initiation and sensorimotor processing is understood. I have no further comments.

Reviewer #2: The authors have adequately addressed all of my previous comments and concerns. I have no further comments.

7. PLOS authors have the option to publish the peer review history of their article (what does this mean?). If published, this will include your full peer review and any attached files.

Reviewer #1: **Yes:**Olgerta Asko

Reviewer #2: No

---

## [Editor Report · Acceptance letter]

PONE-D-25-59260R1

PLOS One

Dear Dr. Branco,

I'm pleased to inform you that your manuscript has been deemed suitable for publication in PLOS One. Congratulations! Your manuscript is now being handed over to our production team.

Kind regards,

on behalf of

Prof. Dr. Onder Aydemir

Academic Editor

PLOS One